# Myosin III-mediated cross-linking and stimulation of actin bundling activity of Espin

**Haiyang Liu[1†], Jianchao Li[2†], Manmeet H Raval[3], Ningning Yao[1], Xiaoying Deng[1], Qing Lu[2], Si Nie[4], Wei Feng[4], Jun Wan[1,2], Christopher M Yengo[3], Wei Liu[1,2*], Mingjie Zhang[1,2,5*]**

[1]Shenzhen Key Laboratory for Neuronal Structural Biology, Biomedical Research Institute, Shenzhen Peking University-The Hong Kong University of Science and Technology Medical Center, Shenzhen, China; [2]Division of Life Science, State Key Laboratory of Molecular Neuroscience, Hong Kong University of Science and Technology, Hong Kong, China; [3]Department of Cellular and Molecular Physiology, Pennsylvania State University College of Medicine, Hershey, United States; [4]National Laboratory of Biomacromolecules, Institute of Biophysics, Chinese Academy of Sciences, Beijing, China; [5]Center of Systems Biology and Human Health, School of Science and Institute for Advanced Study, Hong Kong University of Science and Technology, Hong Kong, China

**Abstract** Class III myosins (Myo3) and actin-bundling protein Espin play critical roles in regulating the development and maintenance of stereocilia in vertebrate hair cells, and their defects cause hereditary hearing impairments. Myo3 interacts with Espin1 through its tail homology I motif (THDI), however it is not clear how Myo3 specifically acts through Espin1 to regulate the actin bundle assembly and stabilization. Here we discover that Myo3 THDI contains a pair of repeat sequences capable of independently and strongly binding to the ankyrin repeats of Espin1, revealing an unexpected Myo3-mediated cross-linking mechanism of Espin1. The structures of Myo3 in complex with Espin1 not only elucidate the mechanism of the binding, but also reveal a Myo3-induced release of Espin1 auto-inhibition mechanism. We also provide evidence that Myo3-mediated cross-linking can further promote actin fiber bundling activity of Espin1.

**\*For correspondence:** liuw@ust.
hk (WL); mzhang@ust.hk (MZ)

[†]These authors contributed equally to this work

## Introduction

Class III myosins (Myo3), together with class IX myosins, are two special groups of the myosin super-family as these two sub-families of actin motors contain enzymatically active domains and thus are regarded as motorized signaling molecules (*Bähler, 2000*). The first member of Myo3 was identified in *Drosophila* photoreceptors and named as NinaC (neither inactivation nor afterpotential C) (*Montell and Rubin, 1988*). There are two paralogs of Myo3 in vertebrate, Myo3a and Myo3b, both of which are known to express in vertebrate retina and cochlea (*Dose and Burnside, 2000*, *2002*; *Shin et al., 2013*). It is believed that they may play partially redundant roles as transporters that are crucial for vertebrate photoreceptor and stereocilia ultrastructure maintenance (*Manor et al., 2012*; *Mecklenburg et al., 2015*; *Merritt et al., 2012*).

Myo3 across different species all contain an N-terminal S/T kinase domain before their motor head. The kinase domain has been reported to regulate the motor's ATPase activity (*Komaba et al., 2010*; *Quintero et al., 2010*). The tail regions of Myo3 from different species are less conserved.

**eLife digest** A mammal's sense of hearing and balance are helped by so-called hair cells within the inner ear. These cells are named after their long, hair-like protrusions called stereocilia, and mutations in the genes involved in stereocilia development and maintenance can lead to hearing loss in humans. Damage to the stereocilia caused by excessive exposure to loud noises can also have the same effect.

Stereocilia are full of filaments made of a protein called actin. Other proteins called class III myosins and Espin are also both required for normal development of stereocilia. Mutations in the genes that encode these proteins can cause hereditary deafness in humans. However, it remains unclear exactly how these two proteins (myosins and Espin) interact with each other in stereocilia.

Using biochemical and structural studies, Liu, Li et al. have now discovered that the so-called 'tail' part of the myosins contains a pair of repeated sequences that can each interact with an Espin protein called Espin1. This interaction allows each myosin to cross-link two Espin1 proteins. Espins assemble actin filaments into bundles, and further experiments showed that this cross-linking interaction between myosins and Espins helped this process, which is linked to stereocilia development and maintenance.

Mammals actually have two related copies of class III myosins that play overlapping but slightly different roles. The next challenge will be to try to understand the differences between these related proteins, as well as to try to uncover the roles of other forms of Espin in stereocilia.

*Drosophila* NinaC contains a PDZ binding motif at its very C-terminus capable of binding to a master scaffold protein called INAD (Inactivation no afterpotential D) (*Wes et al., 1999*). Vertebrate Myo3 tails share a conserved vertebrate specific domain referred to as tail homology I motif (THDI) (*Figure 1A*) (*Dose et al., 2003*). The THDI mediates binding of Myo3 to its cargo protein Espin1 (Ectoplasmic specialization protein 1) and allows Myo3 to transport Espin1 to the tips of actin bundle-based structures such as filopodia and stereocilia. Once tip localized, Espin1 WH2 domain promotes the elongation of actin protrusions (*Merritt et al., 2012*; *Salles et al., 2009*). However, the detailed molecular basis governing the Myo3 and Espin1 interaction is not clear.

Espin1 was first identified in Sertoli cell-spermatid junctions (*Bartles et al., 1996*), encoded by the gene *Espin*. Later, shorter spliced isoforms of *Espin* gene products (Espin2B, Espin3A and Espin4) were shown to be expressed in other F-actin rich structures such as brush border microvilli and Purkinje cell dendritic spines (*Bartles et al., 1998*; *Sekerkova et al., 2003*). They share a common 14 kDa C-terminal actin binding domain (ABD; *Figure 1A*), which was reported to be necessary and sufficient for F-actin bundling activity (*Bartles, 2000*; *Bartles et al., 1998*). Besides the ABD, all Espin isoforms contain a WH2 motif which can bind to actin monomer and a proline rich (PR) region which can interact with profilins (*Sekerkova et al., 2006*). Espin2B contains one more PR region and an extra actin binding site (xAB) at the N-terminus (*Chen et al., 1999*). Espin1 is the longest isoform in the family and contains a stretch of ankyrin repeats (AR) in its N-terminus (*Figure 1A*). The AR of Espin1 is responsible for directly interacting with Myo3 tail THDI (*Merritt et al., 2012*; *Salles et al., 2009*). Recently, it was reported that Espin1 contains an AR binding sequence immediately C--terminal to xAB, and binding of AR to this sequence prevents xAB from binding to actin (*Zheng et al., 2014*). As such, the actin binding activity of the N-terminal part of Espin1 is auto-inhibited, and this AR-binding region is named as the auto-inhibitory region (AI, *Figure 1A*). The authors also proposed that Myo3 binding can release the auto-inhibition and increase the diameter of Espin1-promoted actin bundles (*Zheng et al., 2014*).

Myo3a/Myo3b are known to co-localize with Espin1 at the tips of stereocilia of hair cells (*Merritt et al., 2012*; *Salles et al., 2009*; *Schneider et al., 2006*). Importantly, co-expression of Myo3a and Espin1 can further stimulate elongation of stereocilia in cultured organ of Corti hair cells (*Salles et al., 2009*). When expressed in heterologous cells, Myo3a gets enriched at the tip of filopodia (*Les Erickson et al., 2003*; *Salles et al., 2009*; *Schneider et al., 2006*). Mutations of human *Myo3a* gene is known to cause progressive non-syndromic hearing loss, DFNB30 (*Walsh et al., 2002*). Also, transgenic mice with DFNB30 mutation undergo age-dependent outer hair cell

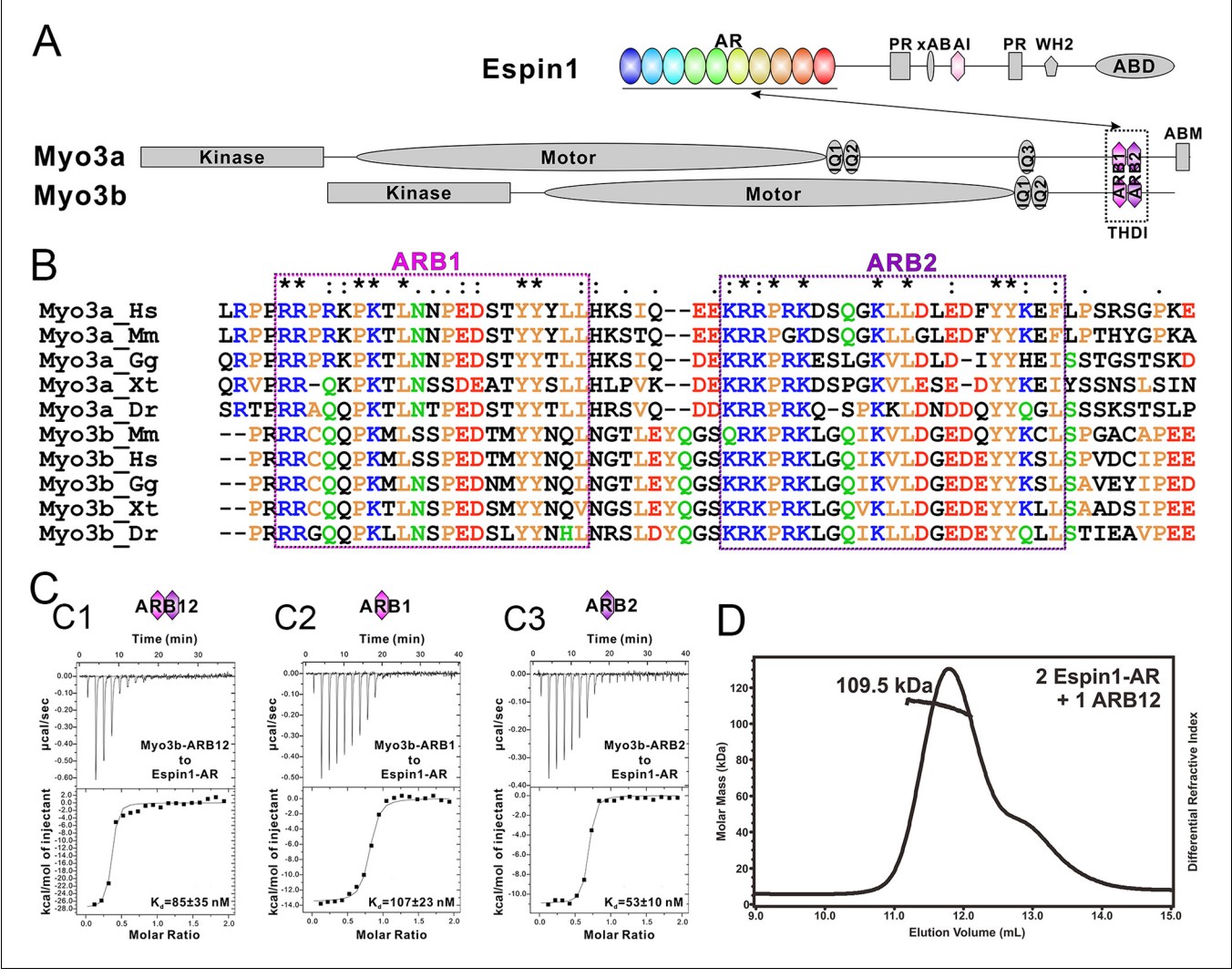

**Figure 1.** Biochemical characterizations of the Myo3/Espin1 interaction. (**A**) Domain organizations of Espin1, Myo3a and Myo3b. (**B**) Sequence alignment of THDI of Myo3a and Myo3b showing that there are a pair of repeating sequences within THDI, which we term as ARB1 and ARB2. Hs, human; Mm, mouse; Gg, chicken; Xt, *Xenopus tropicalis*; Dr, *Danio rerio*. (**C**) ITC results showing that Myo3b-ARB12 (C1) as well as each individual site (C2 for ARB1 and C3 for ARB2) can bind to Espin1-AR with strong affinities. (**D**) FPLC-MALS showing that ARB12 and Espin1-AR form a 1:2 complex.

The following figure supplements are available for figure 1:

**Figure supplement 1.** ITC results of Myo3a-ARBs binding to Espin1-AR.

**Figure supplement 2.** Analytical gel filtration chromatography analysis of the Espin1-AR and Myo3b-ARB12 interaction.

degeneration (*Walsh et al., 2011*). The *jerker* mouse carrying *espin* frame-shift mutation suffers from hair cells degeneration, deafness and vestibular dysfunction (*Sekerkova et al., 2011*; *Zheng et al., 2000*). Mutations in *espin* were also reported to be associated with non-syndromic autosomal recessive deafness DFNB36 and non-syndromic autosomal dominant deafness (*Boulouiz et al., 2008*; *Donaudy et al., 2006*; *Naz et al., 2004*). A prominent phenotype of the *jerker* mice is that their hair cell stereocilia are uniformly thinner and shorter, degenerate faster than those of the wild type littermates (*Sekerkova et al., 2011*; *Zheng et al., 2000*). These genetic findings convincingly point to critical roles of *espin* in stereocilia development and maintenance, likely by promoting the assembly and stabilization of parallel actin filament bundles in stereocilia (*Bartles, 2000*; *Sekerkova et al., 2006*).

Here, we discover that both Myo3a and Myo3b contain two highly similar repeat sequences in their THDI region, each capable of independently binding to Espin1-AR with high affinity. The high resolution crystal structures of each of the two binding sequences from Myo3b in complex with Espin1-AR not only reveal the molecular basis governing the specific interaction between Myo3 and Espin1, but also allow us to discern the Myo3-mediated release of the auto-inhibition mechanism of Espin1. Based on these structural findings, we predict that binding of Myo3 to Espin1 can cluster Espin1 and thus enable Espin1 to further cross-link actin filaments into higher order fibers. Consistent with this prediction, we demonstrate by electron and fluorescence microscopic studies that binding of Myo3 to Espin1 can further promote formation of Espin1-mediated thicker actin bundles.

## Results

### The tail of Myo3 contains two independent Espin1-AR binding repeat sequences

First we analyzed the sequences of the reported Espin1-binding THDI regions of both Myo3a and Myo3b, and found that the region contains a pair of repeating sequences in its N- and C-terminal halves (*Figure 1A and B*, denoted as ARB1 and ARB2 for Espin1 ankyrin repeats binding region 1 and 2 as detailed below). Using isothermal titration calorimetry (ITC)-based quantitative binding assay, we found that purified THDI from both Myo3a and Myo3b can bind to Espin1-AR with high affinities ($K_d$ in the range of tens of nanomolars; *Figure 1C1* and *Figure 1—figure supplement 1A*). Inspection of the ITC-based titration curves indicated that the binding stoichiometry of the Myo3 THDI and Espin1-AR clearly deviates from the value of 1:1 (*Figure 1C1* and *Figure 1—figure supplement 1A*). We thus hypothesized that each of the two repeat sequences in Myo3 THDI may independently bind to Espin1-AR, forming a 2:1 stoichiometric complex. We verified this prediction by gel filtration chromatography and static light scattering experiments (*Figure 1—figure supplement 2* and *Figure 1D*). In the gel filtration analysis, addition of an equivalent molar amount of Espin1-AR to Myo3b-THDI resulted in a complex peak with a smaller elution volume (*Figure 1—figure supplement 2A*). Addition of another molar equivalent of Espin1-AR further shifted the complex peak to a smaller elution volume (*Figure 1—figure supplement 2B*). However, further addition of Espin1-AR did not change the elution volume of the complex any more, indicating that Myo3b THDI is saturated by the binding of two molar ratios Espin1-AR (*Figure 1—figure supplement 2C*). To accurately determine the stoichiometry, we used fast protein liquid chromatography (FPLC) coupled with multi-angle light scattering (FPLC-MALS) to calculate the molecular mass of the Myo3b THDI and Espin1-AR complex. When mixed Myo3b THDI with saturated amount of Trx-tagged Espin1-AR, the fitted molecular weight of the complex peak (~109.5 kDa) matches well with the theoretical molecular weight of 117 kDa for the (Trx-Espin1-AR)$_2$/Myo3b-THDI complex (*Figure 1D*), confirming that Myo3 THDI contains two Espin1-AR binding sites.

Next, we divided Myo3b-THDI into two fragments, each corresponding to ARB1 and ARB2 as shown in *Figure 1B*. Both Myo3 ARB1 and ARB2 bind to Espin1-AR with affinities also in the range of tens of nanomolars and each with a 1:1 stoichiometry (*Figure 1C2 and 1C3*, and *Figure 1—figure supplement 1B and C*), indicating that the two repeating sequences in Myo3-THDI can independently bind to Espin1-AR with comparable affinities. The 1:2 stoichiometry between Myo3 and Espin1 is consistent with a previous finding that human Myo3a THDI lacking exon 33 (exon 33 mainly encodes ARB2) can still interact with Espin1 (*Salles et al., 2009*).

### The overall structure of the Myo3/Espin1-AR complex

To understand the molecular basis of the Espin1/Myo3 interaction, we solved the crystal structure of the Espin1-AR/Myo3b-ARB1 complex at 1.65 Å resolution (*Table 1*). The structure revealed that Espin1-AR contains 10 ANK repeats (*Figure 2*), instead of 8 as predicted earlier (*Bartles et al., 1996*). The repeats 2–9 each contains the signature 'TPLH' sequence at the N-terminus of the αA helix, so can be viewed as the canonical ANK repeats. Like shown in the recently determined structures of the 24 ANK repeats scaffold protein ankyrin-B (*Wang et al., 2014*) and the 9 ANK repeats RNase L (*Han et al., 2012*), the two ANK repeats capping the two termini of Espin1-AR do not contain the 'TPLH' sequence (*Figure 2—figure supplement 1*). We believe that these two non-canonical ANK repeats capping the two termini of Espin1-AR mainly play a structural stabilization role of the

**Table 1.** Statistics of X-ray Crystallographic Data Collection and Model refinement Numbers in parentheses represent the value for the highest resolution shell. a. $R_{merge}=\Sigma\ I_i$- $<I>$ / $\Sigma I_i$, where $I_i$ is the intensity of measured reflection and $<I>$ is the mean intensity of all symmetry-related reflections. b. $R_{cryst}=\Sigma\ F_{calc} - F_{obs}$ /$\Sigma F_{obs}$, where $F_{obs}$ and $F_{calc}$ are observed and calculated structure factors. c. $R_{free}=\Sigma_T F_{calc} - F_{obs}$ /$\Sigma F_{obs}$, where T is a test data set of about 5% of the total unique reflections randomly chosen and set aside prior to refinement. d. B factors and Ramachandran plot statistics are calculated using MOLPROBITY (**Chen et al., 2010**). e. CC* and $CC_{1/2}$ were defined by Karplus and Diederichs (**Karplus and Diederichs, 2012**).

| Data sets | Espin1-AR/Myo3b-ARB1 5ET1 | Espin1-AR/Myo3b-ARB2 5ET0 |
|---|---|---|
| Space group | $P2_1$ | $P2$ |
| Wavelength (Å) | 0.9791 | 0.9795 |
| Unit Cell Parameters (Å) | a=72.74, b=71.14, c=76.88 $\alpha=\gamma=90°$, $\beta=96.88°$ | a=39.74, b=68.78, c=173.45 $\alpha=\gamma=90°$, $\beta=90.04°$ |
| Resolution range (Å) | 50-1.65 (1.68–1.65) | 50-2.30 (2.42–2.30) |
| No. of unique reflections | 93433 (4625) | 39636 (5866) |
| Redundancy | 3.7 (3.7) | 3.7 (3.8) |
| I/$\sigma$ | 18.5 (1.7) | 7.7 (1.9) |
| Completeness (%) | 99.8 (99.9) | 94.9 (96.6) |
| $R_{merge}$[a] (%) | 8.9 (91.6) | 10.3 (79.9) |
| CC* for the highest resolution shell [e] | 0.866 | 0.878 |
| CCi/2 for the highest resolution shell [e] | 0.599 | 0.627 |
| Structure refinement | | |
| Resolution (Å) | 50-1.65 (1.71–1.65) | 10-2.3 (2.38–2.30) |
| Rcryst [b]/Rfree [c] (%) | 16.94/19.11 (25.77/28.64) | 22.32/25.34 (26.74/30.90) |
| rmsd bonds (Å) / angles (°) | 0.006 / 0.795 | 0.010 / 1.113 |
| Average B factor (Å$^2$) [d] | 23.2 | 60.5 |
| No. of atoms | | |
| Protein atoms | 5374 | 4985 |
| Water | 378 | 23 |
| Ligands | 30 | 0 |
| No. of reflections | | |
| Working set | 89061 | 37660 |
| Test set | 4345 | 1925 |
| Ramachandran plot regions [d] | | |
| Favored (%) | 98.9 | 98.4 |
| Allowed (%) | 1.1 | 1.6 |
| Outliers (%) | 0 | 0 |

entire AR fold. The 10 ANK repeats form a left-handed superhelix with the αA helices forming the inner groove and the αB helices forming the outer surface (*Figure 2—figure supplement 2*). Clear additional electron densities lining the inner groove of the ANK repeats allowed us to build the bound Myo3b-ARB1 peptide model with high confidence (*Figure 2A*). The ARB1 binds to Espin1-AR in an antiparallel manner, similar to the binding mode between ANK repeats from Ankyrin R/B/G and their targets (*Wang et al., 2014*) as well as between ANKRA2/RFXANK and their targets (*Xu et al., 2012*), suggesting that elongated inner grooves are common target binding sites of ANK repeats in general. The ARB1 spans nearly the entire inner groove, covering ~1230 Å$^2$ of solvent accessible area. The N-terminal of ARB1 adopts an extended structure and binds to the C-terminal of Espin1-AR. The C-terminal of ARB1 forms an α-helix and binds to the N-terminal half of Espin1-AR (*Figure 2B*). The amino acid sequences of Espin1-AR from different vertebrate species as well as of the mammalian paralogs Espin-like proteins (*Shin et al., 2013*) are highly conserved (*Figure 2—*

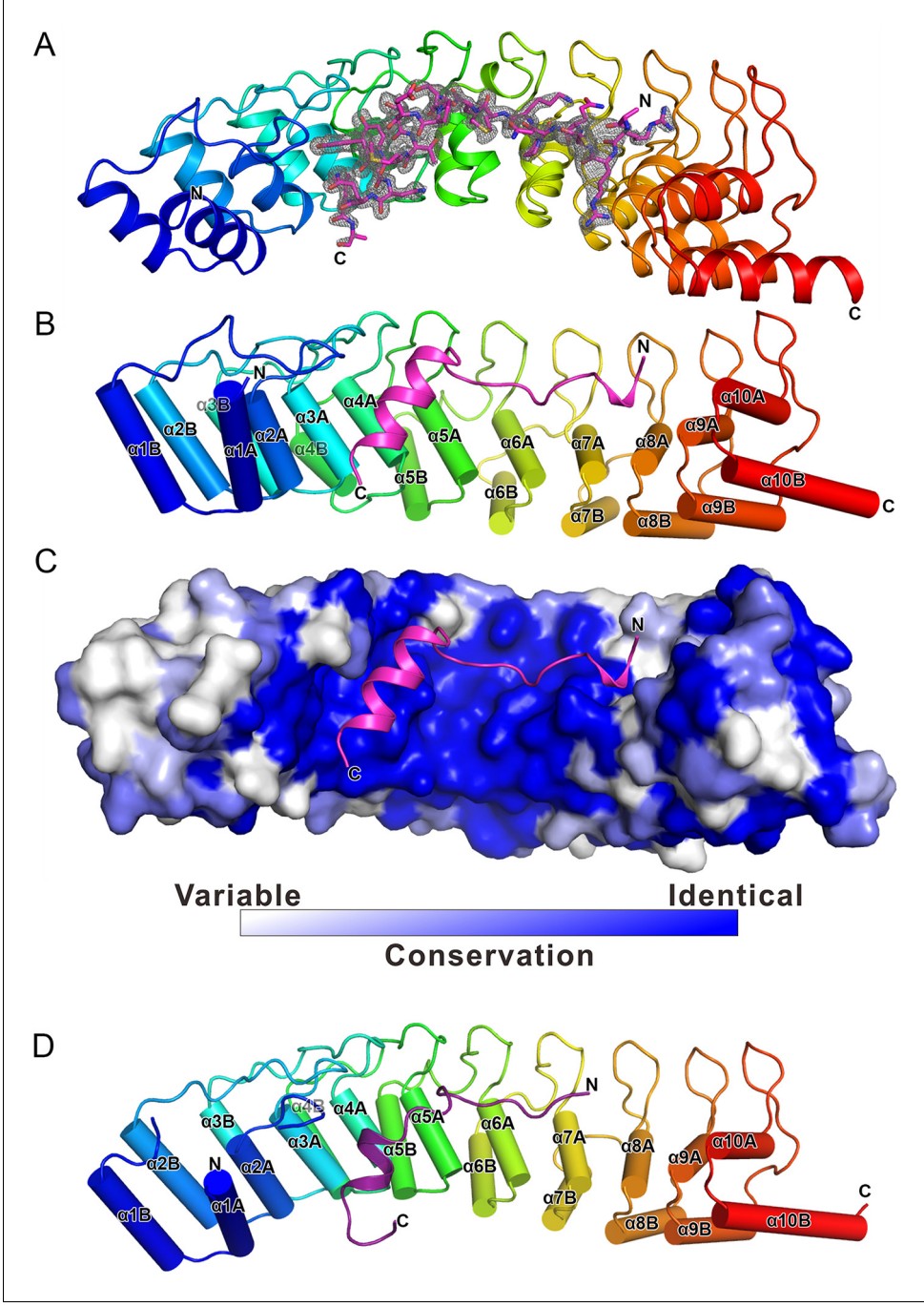

**Figure 2.** The overall structure of the Myo3-ARB/Espin1-AR complex. (**A**) An omit map showing the binding of Myo3b-ARB1 to Espin1-AR. The $F_o$-$F_c$ density map was generated by deleting the Myo3b-ARB1 part from the final model and contoured at 3.0σ. The Myo3b-ARB1 fitting the electron density is displayed in the stick model. (**B**) The overall structure of the Myo3b-ARB1/Espin1-AR complex. The Espin1-AR is shown in cylinders, Myo3b-ARB1 is shown with the ribbon diagram and colored in magenta. (**C**) The amino acid conservation map of Espin1-AR. The conservation map was calculated based on the sequence alignment of vertebrate Espin1 and mammalian Espin-like proteins shown in *Figure 2—figure supplement 1*. The identical residues are colored in dark blue; the strongly similar residues are colored in blue; the weakly similar residues are colored in light blue; the variable residues are colored in white. (**D**) The overall structure of Myo3b-ARB2/Espin1-AR complex. The Espin1-AR is shown in cylinders, Myo3b-ARB2 is shown in ribbon diagram and colored in dark purple.

The following figure supplements are available for figure 2:

*Figure 2 continued on next page*

*Figure 2 continued*

**Figure supplement 1.** Structural-based sequence alignments of AR of Espin1 from different vertebrate species and Espin-like proteins from mammals.

**Figure supplement 2.** Superhelical model of Espin1-AR.

---

*figure supplement 1*). We mapped the sequence conservation profile to the structure of Espin1-AR and found that the residues in the inner groove of AR are highly conserved. In particular the residues in the ARB1 binding surface are essentially totally conserved (*Figure 2C*).

We have also determined the Espin1-AR/Myo3b-ARB2 complex at a resolution of 2.3 Å (*Table 1*). The structure of the complex and the binding mode of Myo3b-ARB2 to Espin1-AR are highly similar to what are observed in the Espin1-AR/Myo3b-ARB1 complex (*Figure 2D*), directly confirming our earlier conclusion that the two repeat sequences in Myo3-THDI bind to Espin1-AR with similar binding mode and affinity. We tried very hard to crystallize the Espin1-AR/Myo3-THDI complexes without success, presumably due to flexibilities of the connection sequences between ARB1 and ARB2 of Myo3.

## The detailed Myo3/Espin1-AR interaction

Since the two complex structures are essentially the same, here we only describe the detailed interactions observed in the Espin1-AR/Myo3b-ARB1 structure, which was resolved at a higher resolution. The Espin1-AR/Myo3b-ARB1 interface can be arbitrarily divided into three regions (*Figure 3A*). The first binding site is formed by the repeats 2–4 of Espin1-AR and binds to the C-terminal α-helix of ARB1 (*Figure 3A1*). Two absolutely conserved tyrosine residues (Tyr1267$_{ARB1}$ and Tyr1268$_{ARB1}$, the double tyrosine ('YY') motif; *Figure 3B*) and Leu1271$_{ARB1}$ in the next turn insert into the hydrophobic pocket in the N-terminal of Espin1-AR (*Figure 3A1*). In addition, the hydroxyl groups of the 'YY' motif also make hydrogen bonds. Mutations of these two tyrosine residues to alanine greatly decreased Myo3b-ARB1's binding to Espin1-AR (*Figure 3C*, *Figure 3—figure supplement 1B*). Similarly, mutation of Leu110 in the Espin1-AR hydrophobic pocket to a polar residue aspartic acid decreased the affinity by ~10-fold (*Figure 3C*, *Figure 3—figure supplement 1C*). Furthermore, the carboxyl group of Asp1264$_{ARB1}$ makes hydrogen bonds with Asn69 and Ser103. Mutation of this residue together with Glu1263$_{ARB1}$ to alanines decreased the affinity by ~10-fold (*Figure 3C*, *Figure 3—figure supplement 1D*). The second region is composed of Espin1 repeats 5–8 and binds to the middle-stretch of ARB1 with an extended conformation (*Figure 3A2*). Both the side chains and backbone carbonyl of Leu1259$_{ARB1}$ are involved in the interaction. Asp205 located in the finger loop between repeat 6 and 7 forms a salt bridge with Lys1257$_{ARB1}$ and a hydrogen bond with Gln1254$_{ARB1}$. Mutation of Lys1257$_{ARB1}$ into a reversed charged residue glutamic acid, together with Leu1259$_{ARB1}$ to alanine substitution also decreased the affinity by about ~10-fold (*Figure 3C*, *Figure 3—figure supplement 1E*). The third region involves inner groove of the Espin1-AR repeats 8–10, which is highly enriched with negatively charged residues (*Figure 3A3*). The two highly conserved Arg residues at the beginning of Myo3b-ARB1 (Arg1251$_{ARB1}$ and Arg1252$_{ARB1}$) insert into the negatively charged pocket (*Figure 3A3*). Mutating these two arginine residues to reverse charged residues glutamic acid decreased Myo3B-ARB1 binding to Espin1-AR by ~15-fold (*Figure 3C*, *Figure 3—figure supplement 1F*). By analyzing the sequence of the ARBs from Myo3, we found that there exist more positively charged residues in addition to the highly conserved Arg residues at the further N-terminal end of ARB2 from both Myo3a and Myo3b (*Figure 3B*). We anticipate that these additional positively charged residues might also be involved in the binding, as there remain unoccupied, negatively charged surfaces in the third region of the Espin1-AR/Myo3b-ARB1 structure (*Figure 3A3*). Indeed, substitutions of the more N-terminal positively charged residues of Myo3b-ARB2 (Arg1282$_{ARB2}$ and Lys1283$_{ARB2}$) with alanines decreased its binding to Espin1-AR by ~10-fold (*Figure 3C*, *Figure 3—figure supplement 1I*).

Comparing the structures of Espin1-AR in complex with Myo3b-ARB1 and Myo3b-ARB2, the 'YY' motif and the 'KxL' motif are essentially in the same places (*Figure 3—figure supplement 2B and C*). Despite the high similarity, there are still a few minor differences. First of all, the C-terminal α-helix of ARB2 is shorter. The interaction is mediated by a hydrogen bond between Asp1294 and

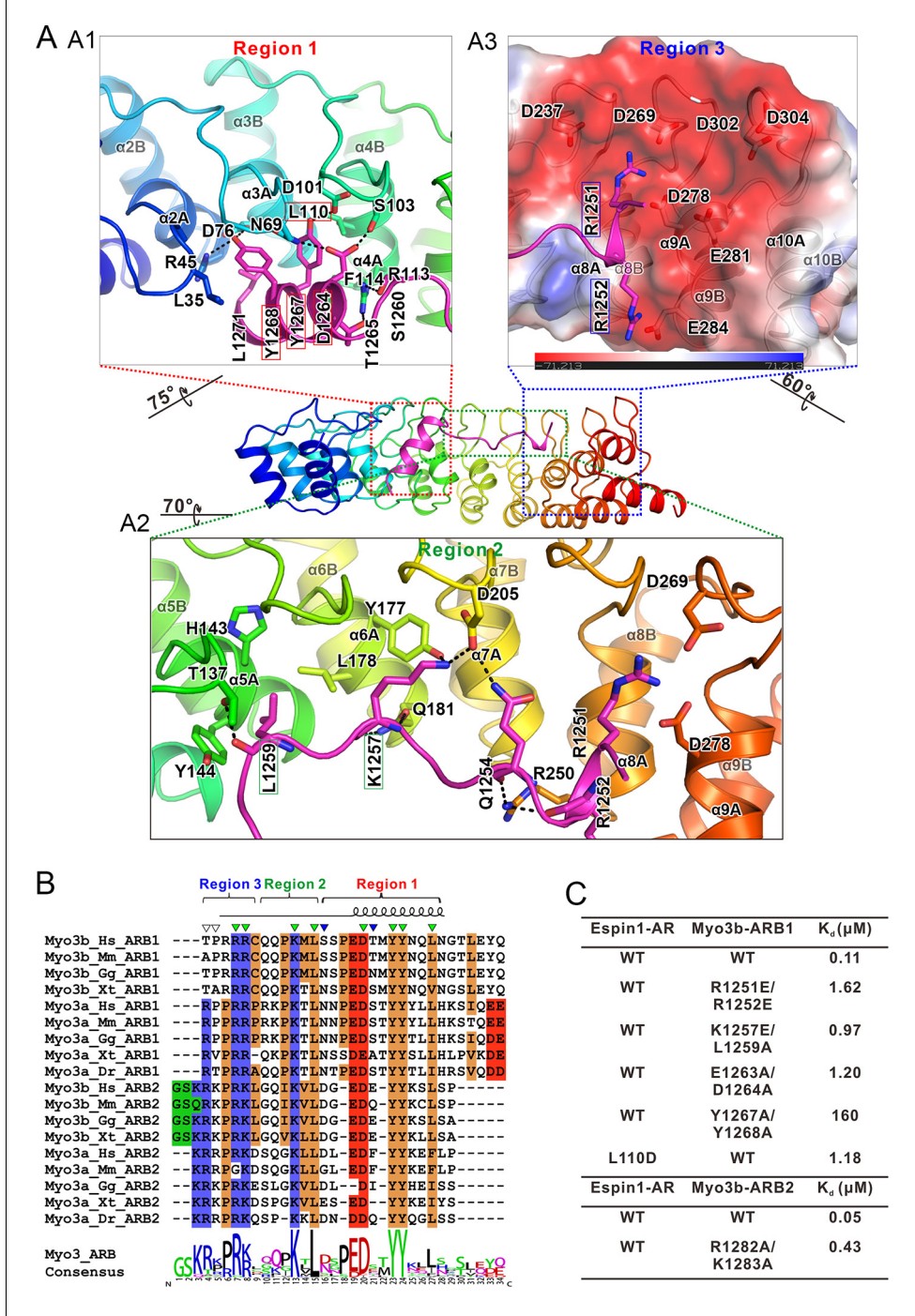

**Figure 3.** The detailed Myo3/Espin1-AR interaction.  (**A**) The Myo3b-ARB1/Espin1-AR interface is divided into three regions corresponding to the 'YY' motif (A1), the 'KxL' motif (A2) and the N-terminal positively charged residues (A3) of Myo3b-ARB1. The residues tested with the mutagenesis experiments are highlighted with boxes. The side chains or main chains of the residues involved in the interactions are highlighted in the stick model. Charge-charge and hydrogen bonding interaction are highlighted by dashed lines. The electrostatic surface potentials were calculated using PyMol. (**B**) Sequence alignment of Myo3-ARBs showing the conservation of ARBs. The conserved residues involved in the binding are highlighted with solid green triangles. The variable residues involved in the binding are highlighted with solid blue triangles. The two positively charged residues in ARB2 that are not resolved in the structure are highlighted with unfilled triangles. The sequence logo beneath the alignment was generated using WebLogo (*Crooks et al., 2004*). (**C**) ITC derived dissociation constants showing that

*Figure 3 continued on next page*

*Figure 3 continued*

mutations of the critical residues in the interface invariably weakened the binding. The original ITC data are shown in *Figure 3—figure supplement 1*.

The following figure supplements are available for figure 3:

**Figure supplement 1.** The ITC titration curves for calculating the dissociation constants shown in *Figure 3C*.

**Figure supplement 2.** Comparison of Myo3b-ARB2/Espin1-AR and Myo3b-ARB1/Espin1-AR structures.

Tyr144, instead of the more extensive interaction observed in Myo3b-ARB1 complex (*Figure 3—figure supplement 2B*). Moreover, the positively charged residues in the N-terminus of ARB2 cannot be reliably built, probably due to the high salt concentration in the crystallization buffer (1.6 M ammonium sulfate). Nonetheless, clear electron density can be observed near the negatively charged surface (*Figure 3—figure supplement 2A*). Indeed, substitutions of these positively charged residues with Ala weakened the binding (*Figure 3C* and *Figure 3—figure supplement 1G–I*). Furthermore, the involvement of more positively charged residues of ARB2 may compensate for the less extensive interaction in its shorter C-terminal helix, thus resulting in a similar binding affinity to Espin1-AR as ARB1 does (107 nM for ARB1 vs 53 nM for ARB2, *Figure 1C2 and C3*).

## Binding of Myo3 releases the auto-inhibition of Espin1

It was reported that a conserved region following the xAB segment of Espin1 can interact with the N-terminal AR (*Figure 4A*) and inhibit the actin binding activity of xAB (*Zheng et al., 2014*). By comparing the sequence of AI (aa 496–529) with the consensus sequence of Myo3-ARBs, we find that Espin1-AI bears high sequence homology with Myo3-ARBs (e.g. the completely conserved 'YY' motif, the central 'KxL' motif, and the N-terminal positively charged residues; *Figure 4B*). Thus, we predict that AI may bind to Espin1-AR with a similar binding mode as Myo3-ARBs do. We used ITC-based binding assay to test this prediction, and found that Espin1-AI can indeed bind to Espin1-AR, albeit with a more moderate affinity than Myo3-ARBs ($K_d$ of 1.32 μM vs 0.05~0.1 μM) (*Figure 4C1*). Fully consistent with our structure-based sequence alignment analysis, substitutions of the two tyrosines in Espin1-AI to alanines greatly weakened its binding to Espin1-AR (*Figure 4C2*). Given that the AI segment (aa 496–529) is immediately C-terminal to xAB (aa 462–487) of Espin1 (*Figure 4A*), one might envision that the interaction between Espin1 AR and AI can conformationally mask the xAB's actin binding activity and thus renders Espin1 in an auto-inhibited conformation.

It was shown that a synthetic peptide encompassing the Myo3a-ARB1 sequence identified here can stimulate the actin binding activity of xAB (*Zheng et al., 2014*). Based on our analysis, the most likely mechanism for Myo3a/b-ARB1-mediated stimulation of xAB's actin binding may be due to the release of AI binding from Espin1-AR by direct competition of Myo3a/b-ARB1 binding. We designed biochemistry experiments to support the above model. If Myo3-ARB1 can indeed compete with AI for binding to Espin1-AR, then Myo3-ARB1 must still be able to bind to the auto-inhibited Espin1 but with an affinity weaker than binding to the isolated Espin-1-AR. We obtained highly purified N-terminal auto-inhibitory fragment of Espin1 spanning from AR to AI (denoted as Espin1-1-529, *Figure 4A*) and the full length Espin1 (denoted as Espin1-FL), and found that both proteins exist as monomer in solution (*Figure 4—figure supplement 1*), indicating that Espin1 auto-inhibition is intramolecular in nature. ITC-based assay further showed that Myo3a-ARB1 can indeed bind to both Espin1-1-529 and Espin1-FL and with a weaker binding affinity than binding to Espin1-AR (*Figure 4D1–3*), consistent with a partially blocked Espin1-AR binding groove by AI. We also noticed that the ITC titration reactions of Myo3a-ARB1 to Espin1-1-529 and Espin1-FL are endothermic (*Figure 4D1 and 4D2*) instead of the exothermic reactions between Myo3a-ARB1 titrating to Espin1-AR (*Figure 4D3*), further indicating that the binding of Myo3a-ARB1 to the auto-inhibited Espin1 is not a simple direct association process between ARB1 and AR. To provide further proof, we truncated Espin1 from the C-terminus just before the AI (i.e., aa 1–494, denoted as Espin1-1-494) and found that Espin1-1-494 binds to Myo3a-ARB1 with an affinity similar to that between Espin1-AR and Myo3a-ARB1 (*Figure 4D3*), indicating that AI is indeed responsible for the decreased binding of Espin1 to Myo3a-ARB1.

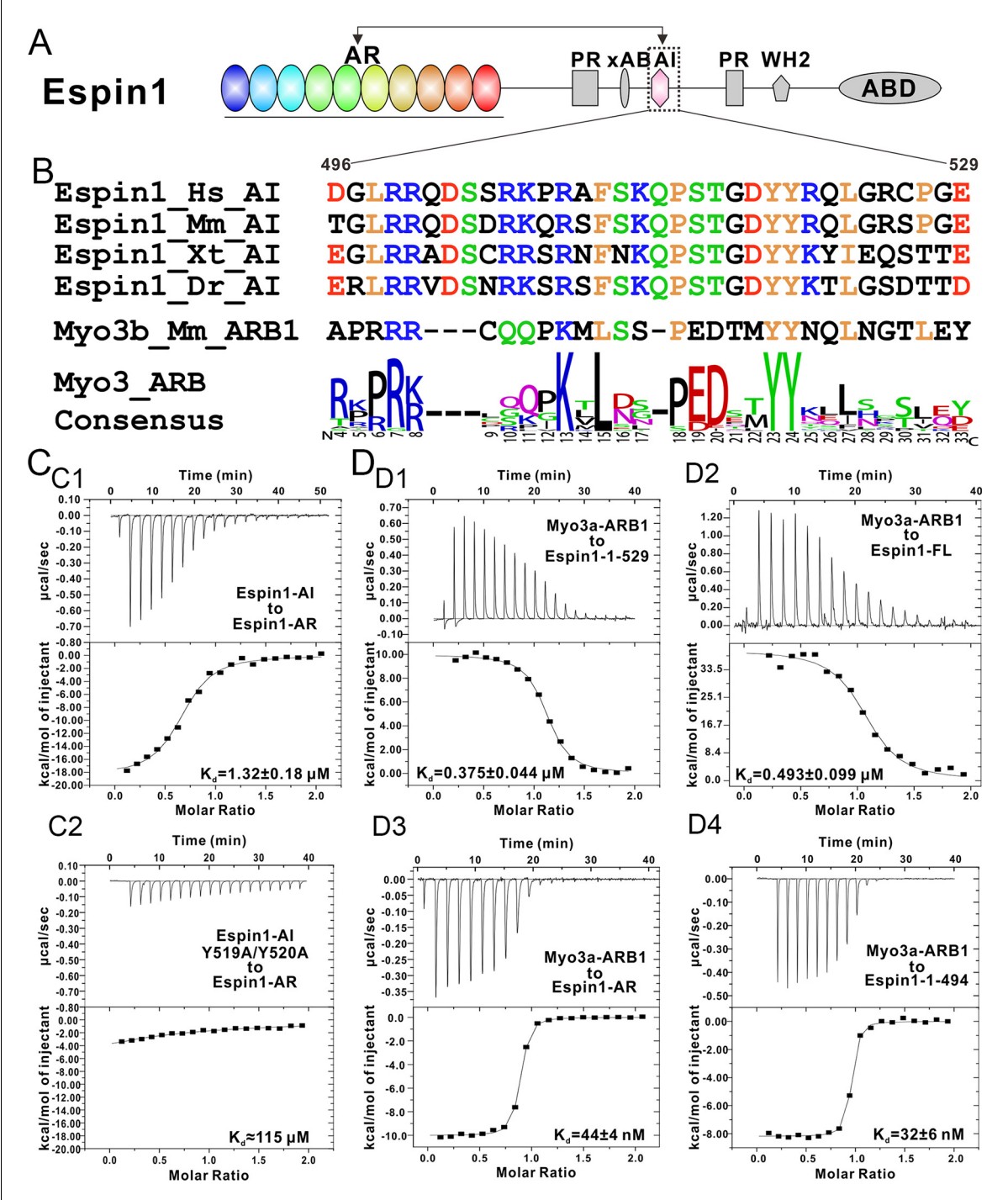

**Figure 4.** Biochemical characterization of the Espin1 auto-inhibition. (A) Domain organization of Espin1 showing that the Espin1-AI in the middle may bind to Espin1-AR at the N-terminus. (B) Sequence alignment of Espin1-AI from different vertebrate species, and comparison of Espin1-AI with the consensus sequence of Myo3-ARBs as shown in *Figure 3D*. (C) ITC result showing that Espin1-AI binds to Espin1-AR with a moderate affinity (C1). Mutation of the 'YY' motif to alanine greatly decrease the binding (C2). (D) ITC results showing that Myo3a-ARB1 can still bind to Espin1-1-529 (D1) and Espin1-FL (D2) with a sub-micromolar affinity. In contrast, Myo3a-ARB1 binds to Espin1-AR (D3) and Espin1-1-494 with comparable strong affinities (D4). Panel D3 is the same as *Figure 1—figure supplement 1B*.

The following figure supplement is available for figure 4:

**Figure supplement 1.** FPLC-MALS results of Espin1-1-529 and Espin1-FL.

## Espin1 binding sites in Myo3 are critical for the filopodia tip localization of Espin1 and Myo3

Myo3a is known to localize at the tip of filopodia when transfected in heterologous cells like HeLa or COS7 cells, whereas Myo3b alone cannot tip-localize as it lacks ABM (*Les Erickson et al., 2003*; *Salles et al., 2009*). However, when co-expressed with Espin1, Myo3b can bind to Espin1 and localize to the tip of filopodia (*Manor et al., 2012*; *Merritt et al., 2012*). Similarly, Myo3a lacking ABM can only tip-localize when co-expressed with Espin1. Deletion of the Myo3 kinase domain is known to render the motor in a constant active state in promoting the length of filopodia, thus we used Myo3 constructs lacking the kinase domain in the subsequent experiments (*Les Erickson et al., 2003*; *Quintero et al., 2010*; *2013*). We first tested the role of Myo3/Espin1 binding on Myo3a's ability to tip-localize. To test our biochemical findings and to determine the impact of Myo3-ARB 'YY' motifs on Myo3-Espin1 interaction, Espin1 transportation (i.e., tip localization) and filopodia elongation, we co-expressed various Myo3aΔKΔABM (lacking the kinase domain and the ABM) and Myo3bΔK constructs with Espin1 in COS7 cells. Since both Myo3aΔKΔABM and Myo3bΔK cannot tip-localize by its own, we reasoned that when co-expressed with Espin1, the Myo3 and Espin1 tip localization levels will determine the intactness of their mutual binding. As expected, both GFP-tagged wild type Myo3a (lacking the kinase domain and the ABM, denoted as ΔKΔABM) and RFP-tagged Espin1 localized to the tip of filopodia when co-transfected in COS7 cells (*Figure 5A1*). Mutation of the 'YY' motif of either of the ARBs (denoted as mARB1 and mARB2) only had a moderate or even unobservable effect on filopodia tip localization for both Myo3a and Espin1 (*Figure 5B*), indicating that ARB1 and ARB2 may play some redundant functions in this overexpression system. Mutations of both 'YY' motifs (mARB1+mARB2, denoted as mARB) significantly reduced the filopodia tip localization of Myo3a (*Figure 5A2* and *Figure 5B*). Similarly, deletion of one ARB had a moderate effect and deletion of both ARBs had a much more severe effect on filopodia tip localization of Myo3a (*Figure 5A3* and *Figure 5B*). We have also tested the effect of the corresponding set of mutations or deletions of ARBs on Myo3b, and observed similar results (*Figure 5C* and *Figure 5—figure supplement 1*) as those of Myo3a. It is worth noting that the ratio of tip to cell body protein level of Myo3b is significantly less than that of Myo3a (*Figure 5B and C*), which is also consistent with previously reported findings (*Manor et al., 2012*; *Merritt et al., 2012*), but the detailed mechanism for this difference is unknown. The above results demonstrate that both ARBs in Myo3a or Myo3b are important for filopodia tip localizations of Myo3 and Espin1.

## Myo3 binding promotes Espin1's higher order actin bundling activity

Our above structural and biochemical characterizations of the Myo3 and Espin1 interactions point to a likely regulatory role of Myo3 on Espin1's actin binding and bundling activity. It can be deduced that binding of Myo3-ARBs can first release the auto-inhibited conformation of Espin1. Perhaps more importantly, formation of Myo3/Espin1 complex leads to two Espin1 molcules to be juxtaposed to each other, forming a Myo3 cross-linked Espin1 dimer. Due to the large space between the N-terminal AR and C-terminal ABD in Espin1 (*Figure 1A*), this Myo3-mediated cross-linking positions the two copies of Espin1 ABD at a distance considerably larger than those allowed by other known actin cross-linking proteins. Therefore, we predicted that binding of Myo3 may stimulate higher order actin bundling activity of Espin1. We used both fluorescent microscopy (FM) and transmission electron microscopy (TEM) techniques to exam the Espin1-mediated F-actin bundles with and without the binding of Myo3 ARBs (*Figure 6*). Under FM, the F-actin alone showed only background signal when probed by fluorescence-labeled phalloidin since individual F-actin is too small to be resolved (*Figure 6B*, left). This is consistent with a previous report (*Zheng et al., 2014*), and also directly revealed by our TEM study showing the nm sized F-actin filaments (*Figure 6B*, right). When Espin1 was added into the F-actin solution, uniform needle like bundles could be observed under FM. TEM showed that the bundle size is ∼190 ± 8 nm in width (mean ± SEM) (*Figure 6A1 and C*, *Figure 6—figure supplement 1A*). According to the auto-inhibited model, these actin bundles are probably induced by the C-terminal ABD of Espin1. By adding ARB1 or ARB2 to Espin1 and F-actin containing solution, some of the actin bundles were cross-linked and formed clusters under FM and TEM (*Figure 6A2 and 3*). Similar phenomena have also been observed earlier using Myo3/Espin1/actin co-polymerization bundling assay (*Zheng et al., 2014*), instead of the post-polymerization bundling assay employed in this study. From the TEM images, the diameter of the cluster is slightly

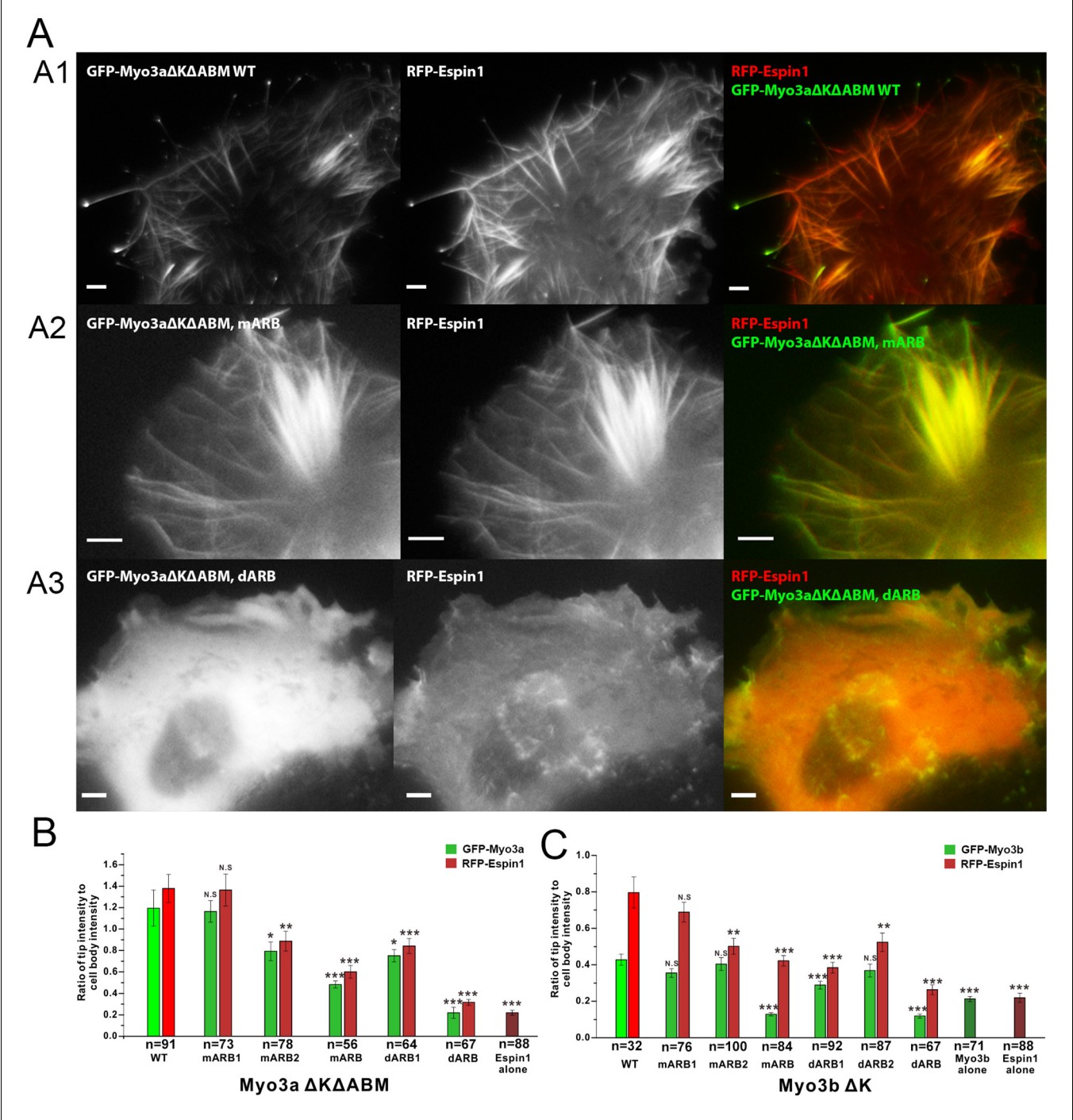

**Figure 5.** Myo3-ARBs/Espin1 interaction is critical for the filopodia tip localizations of Espin1 and Myo3. (**A**) Representative fluorescence images of COS7 cells co-expressing RFP-Espin1 and various GFP-Myo3a experimental constructs. A1, Myo3aΔKΔABM WT; A2, Myo3aΔKΔABM mARB; A3, Myo3aΔKΔABM dARB. Scale bar: 5 μm. (**B**) Quantifications of the tip to cell body ratios of GFP-Myo3a (or its mutants) and RFP-Espin1 based on the experiments shown in panel A. (**C**) Quantifications of the tip to cell body ratios of GFP-Myo3b (or its mutants) and RFP-Espin1 when expressed in COS7 cells. The representative images for this group of experiments are shown in *Figure 5—figure supplement 1*. Values are means ± SEM and analyzed with Two-tailed Student's *t* test; *p<0.05, **p<0.01, ***p<0.001.

The following figure supplement is available for figure 5:

**Figure supplement 1.** ARBs are required for both Myo3b and Espin1 filopodia tip localization.

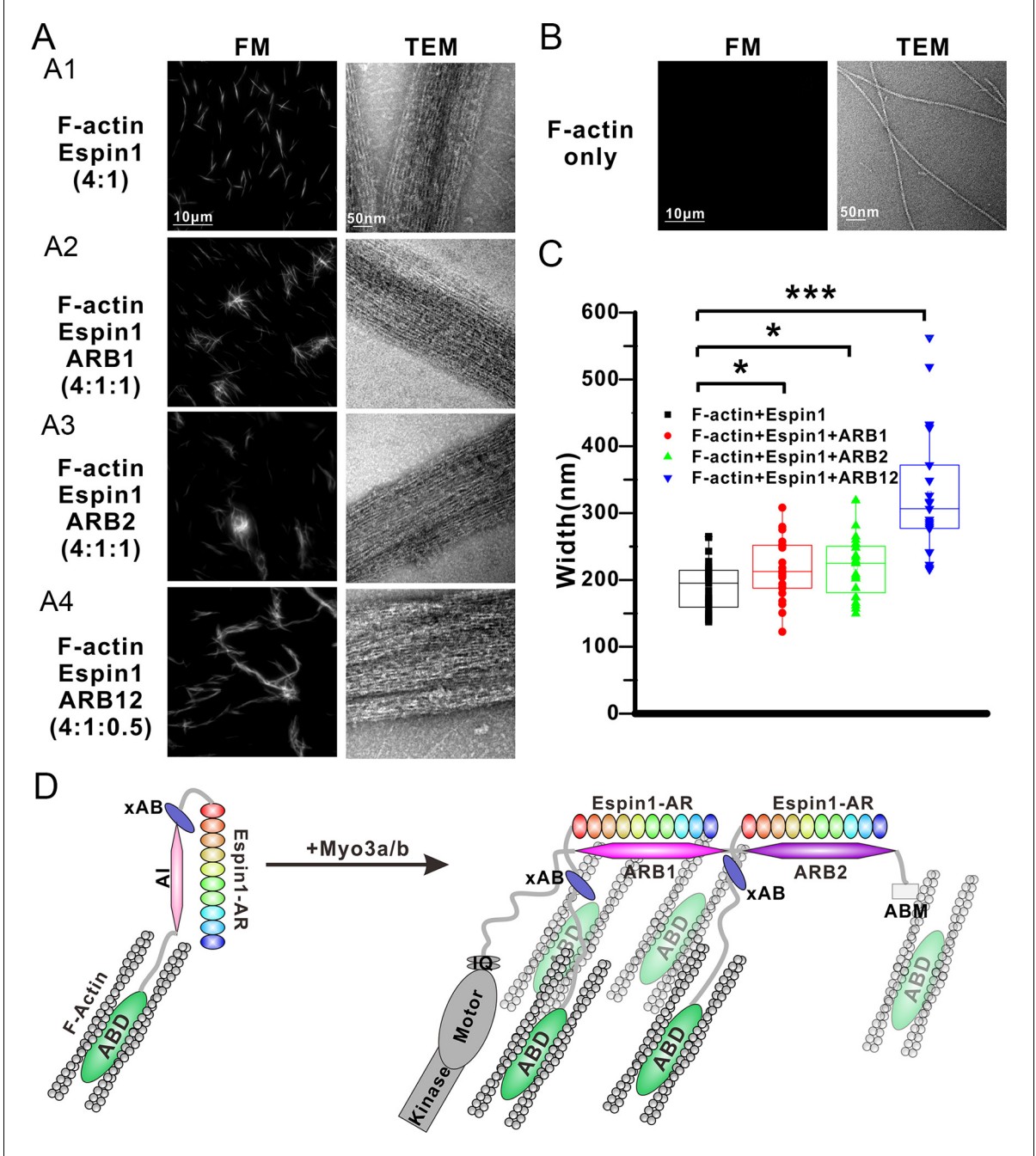

**Figure 6.** Myo3 binding promotes Espin1's higher order actin bundling activity. (**A**) Representative images of actin bundles induced by Espin1 with and without the presence of various forms of Myo3-ARBs under fluorescent microscopy (left) or transmission electron microscopy (right). A1: F-actin+Espin1 with a 4:1 molar ratio mixing; A2: F-actin+Espin1+ARB1 with a 4:1:1 molar ratio mixing; A3: F-actin+Espin1 +ARB2 with a 4:1:1 molar ratio mixing; A4: F-actin+Espin1+ARB12 with a 4:1:0.5 molar ratio mixing. The scale of each column is the same and is indicated at the top panel. (**B**) Representative images of F-actin only under fluorescent microscopy (left) and transmission electron microscopy (right). (**C**) Distribution of the width of actin bundles from the different groups of experiments. Black: F-actin+Espin1; red: F-actin+Espin1+ARB1; green: F-actin+Espin1+ARB2; blue: F-actin+Espin1+ARB12. Statistics are performed by box plot as well as Two-tailed Student's *t* test. *p<0.05, **p<0.01, ***p<0.001. (**D**) Cartoon diagram showing Myo3 mediated higher order actin bundling by Espin1. Without Myo3, the thin actin bundles were induced by Espin1-ABD (left). With Myo3, the two ARBs can bring two Espin1 together, facilitating the formation of higher order actin bundles (right). The xAB region of this Myo3-activated Espin1 and the ABM of Myo3a may further stabilize the higher order actin bundles.

The following figure supplement is available for figure 6:

*Figure 6 continued on next page*

*Figure 6 continued*

**Figure supplement 1.** Representative TEM images of actin bundles.

larger than Espin1 only (*Figure 6A2 and A3*, *Figure 6—figure supplement 1B and C*). Quantification of the F-actin bundle width from the TEM images revealed that the addition of ARBs slightly increased the width of the actin bundles, with 216 ± 9 nm for the ARB1 group and 218 ± 9 nm for the ARB2 group (*Figure 6C*). The above data suggest that the cluster is formed by the Espin1-ABD mediated actin bundles and the ARB1/ARB2 binding releases the xAB, providing an additional actin binding site for forming slightly wider and more branched actin bundles. Interestingly, when ARB12 were added to Espin1 and F-actin containing solution, significantly thicker, elongated, and less branched bundles were observed under FM (*Figure 6A4*, left), a morphology clearly distinct from that with the addition of either ARB1 alone or ARB2 alone. When examined under TEM, the thicker bundles appear to be composed of several thinner actin bundles as observed in the Espin1-ABD-promoted actin bundles (*Figure 6A4*, right and *Figure 6—figure supplement 1D*), indicating that Myo3-ARB12-mediated binding of Espin1 can promote/stabilize parallel actin fiber formation presumably due to Myo3 mediated cross-linking of Espin1. Quantification of the width of actin bundles formed in the presence the Myo3-ARB12 and Espin1 showed that the average width of the actin bundles are significantly larger (330 ± 20; with the thickest bundles reaching ~600 nm in width) than the other three groups (*Figure 6C*). As a control, Myo3a-ARB12 alone could not cause bundling of F--actin (*Figure 6—figure supplement 1E*). Based on these results, we propose a model that, first the Espin1-ABD is able to bundle F-actins, forming the thin F-actin bundles; then the two Espin1 binding sites located at the tail of Myo3 not only release the auto-inhibition of but also cross-link Espin1, further assembling the thin actin-bundles into thicker parallel actin bundle fibers (*Figure 6D*). Therefore, one can envision that a key function of the Myo3/Espin1 interaction is to promote formation of thicker parallel actin bundle fibers in cellular structures such as stereocilia.

## Discussion

The most important findings of this study are the structure-based discovery of two strong and independent Espin1 binding sites in the tail region of each class III myosins, and Myo3 binding-induced auto-inhibition release of Espin1. These discoveries, together with our cell-based filopodia formation and localization assay and microscopic-based actin bundling assay, provide compelling evidences showing that Myo3 and Espin1 work together to assemble and promote higher order parallel actin bundle formation in cellular processes such as stereocilia. We believe that our model reveals the probable underlying molecular mechanism of hearing loss development (stereocilia degeneration) in humans with mutations in Myo3a (DFNB30) (*Walsh et al., 2002*). It is believed that the delayed hearing loss phenotype in DFNB30 subjects could be due to Myo3b compensatory mechanism (*Manor et al., 2012*), a hypothesis supported by our results as well. With the help of the motor domain-mediated high affinity binding of Myo3 to actin filaments (*Dose et al., 2007*; *Kambara et al., 2006*), the full-length Myo3/Espin1 complex may have even higher actin fiber bundling activity than the Myo3-ABR12/Espin1 complex studied in this work. As such, our study reveals a previously unrecognized direct myosin binding-induced actin bundling activity regulation mechanism of an actin filament cross-linker protein. It also demonstrates an intimate synergistic action mechanism between two classes of actin binding proteins, namely actin filament-based myosin motors and actin filament cross-linking protein, in controlling actin fiber size and stability. It has been reported that another unconventional myosin in stereocilia, Myosin XVa, can form a complex with an actin capping protein Eps8 to regulate stereocilia elongations (*Manor et al., 2011*), suggesting that working together with actin binding proteins to regulate actin skeletal dynamics might be a new mode of function for unconventional myosins.

Fitting with this model, stereocilia in *jerker* mice are thinner in their diameters likely due to loss of Espin1-mediated higher order actin fiber assembly, easier to form tapered structures and easier to degenerate presumably due to stability decrease of less bundled actin fibers in stereocilia (*Sekerkova et al., 2011*; *Zheng et al., 2000*). As one of the three known actin cross-linkers in stereocilia, the abundance of Espin is relatively low compared to fascin and plastin (*Shin et al., 2013*). It

has been proposed that Espin may play a regulating role in elongation and widening of actin bundle by concentrating at the tip of stereocilia instead of major structural cross-linking roles (*Avenarius et al., 2014*; *Loomis et al., 2003*). Interestingly, the abundance of Myo3 (both Myo3a and Myo3b together) is similar to that of Espin (a few hundred copies per stereocilium) (*Shin et al., 2013*), indicating that Myo3 and Espin1 may work together to regulate higher order actin bundle structure formation and stability in stereocilia. Both Myo3 and Espin1 are known to concentrate at the tips of stereocilia in hair cells and at the tips of filopodia when expressed in heterologous cells (*Merritt et al., 2012*; *Salles et al., 2009*; *Schneider et al., 2006*). It is possible that stabilization of the growing end of the actin filaments at the plus ends by both Myo3 and Espin1 is a critical driving force for actin filament elongations (*Avenarius et al., 2014*; *Loomis et al., 2003*; *Shin et al., 2013*).

The Myo3 binding-mediated cross-linking of Espin1 cargo is in sharp contrast to the cargo binding-mediated motor dimerization and processivity induction known in a number of other unconventional myosins (myosin V, VI and VII) (*Lu et al., 2014*; *Sakai et al., 2011*; *Shi et al., 2014*; *Yu et al., 2009*). Such large differences in their cargo bindings probably match with the distinct functional properties of these myosins. For a fast-moving cargo transporting motor such as myosin VI, it is important that the motor assumes as a dimer and gets activated for moving in one direction upon binding to a cognate cargo. In such a case, the binding of a cargo protein to the motor tail exerts large impact on the motor activity regulations. As a high duty ratio motor, processive and rapid movement along the actin filaments is not likely to be the main function of Myo3, although the motor can still move towards the plus end of actin filaments (*Les Erickson et al., 2003*; *Merritt et al., 2012*; *Salles et al., 2009*; *Schneider et al., 2006*). Instead, one of Myo3's key functions appears to coordinate with its cargo protein Espin1 to regulate Espin1-mediated actin filaments assembly and stability. Whether such myosin binding-mediated cargo activity regulation is also adopted by other unconventional myosins, particularly for those not known to play transporting roles, is an interesting research topic in the future.

We provide detailed structural information regarding Myo3b/Espin1 interaction. We demonstrate that the Espin1 binding sites in Myo3a and Myo3b are essentially identical and therefore are predicted to be functionally interchangeable. These findings provide a molecular explanation for the partially redundant functions of Myo3a and Myo3b in hair cells. This analysis also predicts that Espin-like proteins can also bind to Myo3 as Espin1 does, suggesting that the functions of Espin1 and Espin-like protein may also be partially redundant in tissues like stereocilia (*Shin et al., 2013*). Such redundancies suggest that mutation of single *myo3a* or *myo3b* or defects in *espin1* or *espin-like* may not always develop severe phenotypes in vertebrates. Moreover, we note that *forked*, the *Drosophila* ortholog of *Espin*, is expressed in bristle cells, a cell type that is enriched in bundled actin filaments (*Petersen et al., 1994*). *Drosophila* Forked protein is predicted to contain five ANK repeats, corresponding to Espin1 repeats 5–9. The structures solved here of mammalian Espin1-AR should help for identifying potential binding partners of Forked-AR.

Although Myo3a and Myo3b are highly similar, there are clear differences in terms of filopodia tip localizations (*Manor et al., 2012*; *Merritt et al., 2012*). Myo3a localize to the very tip of filopodia, while Myo3b distributes in a wider tip-base gradient (*Merritt et al., 2012*). The different tip localization patterns of Myo3a and Myo3b does not originate from the unique ABM of Myo3a (*Manor et al., 2012*). Our study here suggests that their Espin1 binding THDI regions are not responsible for different localizations of Myo3a and Myo3b along stereocilia/filopodia either. It is possible that the kinetics and the F-actin binding affinities of their motor domains or other variable regions in their tails may contribute to such differences.

Most of the unconventional myosins use their globular cargo binding domains to recognize their cargoes (*Lu et al., 2014*). However, Myo3 use their unstructured, yet highly conserved tails to recognize globular domains from their cargoes. Myo3 and another unconventional myosins, Myosin XIX, are not predicted to contain folded globular domains in their tail regions (*Lu et al., 2014*). Based on the results in this study, we anticipate that these two myosins might use different unstructured fragments in their tails to specifically recognize various cargoes. A very recent report showed that another region in Myo3a's tail, which is located immediately N-terminal to the Espin1 binding ARB12 region, can bind to MORN4 (*Mecklenburg et al., 2015*), the mammalian ortholog of *Drosophila* Retinophilin. It is noted that Retinophilin can also bind to *Drosophila* NinaC (*Venkatachalam et al., 2010*). Future studies are required to elucidate the molecular basis

governing the Myo3/MORN4 and NinaC/Retinophilin interactions for better understanding of the interactions.

## Materials and methods

### Constructs and protein expression

The coding sequences of Myo3b-ARB12 (Accession Number: NP_796350.2, aa 1234–1333) and Espin1-AR (Accession Number: NP_997570.1, aa 1–352) were PCR amplified from mouse cDNA library. The full-length human Myo3a and Espin1 plasmids have been described earlier (*Merritt et al., 2012*; *Salles et al., 2009*). The mouse Myo3b-ARB1 (aa 1234–1279), Myo3b-ARB2 (aa 1280–1333), Espin1-AR (aa 1–352), human Myo3a-ARB1 (aa 1488–1520), Myo3a-ARB2 (aa 1521–1553), Espin1-AR (aa 1–352), Espin1-AI (aa 496–529), Espin1-1-494, Espin1-1-529 and Espin1-FL were cloned into an in-house modified pET32a vector (*Liu et al., 2011*). The mouse Myo3b-ARB12 and human Myo3a-ARB12 (aa 1488–1553) were cloned into a pETM.3C vector. All truncations and point mutations of Myo3 and Espin1 used in the current study were created with the standard PCR-based mutagenesis method and confirmed by DNA sequencing. For heterologous cell expressions, the full-length human Myo3a and deletions or mutations were cloned into a modified EGFP vector and the full-length human Espin1 was cloned into a modified RFP vector.

All proteins were expressed in *Escherichia coli* BL21 (DE3) except for Myo3a-ARB12 and Myo3a-ARB2 which were expressed in *Escherichia coli* Rosetta (DE3). The N-terminal thioredoxin-His$_6$-tagged or His$_6$-tagged proteins were purified with a Ni Sepharose 6 Fast Flow column and subsequent Superdex-200 prep grade size-exclusion chromatography.

### FPLC coupled with multi-angle light scattering

Protein samples (typically 100 μl at a concentration of 50 μM pre-equilibrated with column buffer) was injected into an AKTA FPLC system with a Superose-12 10/300 GL column (GE Healthcare) using the column buffer of 50 mM Tris-HCl (pH 7.8), 1 mM DTT, 1 mM EDTA, and 100 mM NaCl. The chromatography system was coupled to a multi-angle light scattering system equipped with a 18 angles static light scattering detector (Dawn, Wyatt) and a differential refractive index detector (Optilab, Wyatt). The elution profiles were analyzed using the ASTRA 6 software (Wyatt).

### Crystallography

Crystals of the Espin1-AR/Myo3b-ARB1 complex and Espin1-AR/Myo3b-ARB2 complex (both in 50 mM Tris-HCl, pH 7.8, 100 mM NaCl, 1 mM EDTA, 1 mM DTT buffer) were obtained by sitting drop vapor diffusion methods at 16°C. The crystals of the Espin1-AR/Myo3b-ARB1 complex were grown in buffer containing 0.2 M lithium acetate and 20% w/v PEG3350 and soaked in crystallization solution containing additional 25% glycerol for cryoprotection. The crystals of Espin1-AR/Myo3b-ARB2 complex were grown in buffer containing 1.6 M ammonium sulfate, 0.1 M Tris pH 8.0 and soaked in crystallization solution containing higher concentration of ammonium sulfate for cryoprotection. Diffraction data were collected at the Shanghai Synchrotron Radiation Facility BL17U at 100 K. Data were processed and scaled using HKL2000 (*Otwinowski and Minor, 1997*).

Structure of the Espin1-AR/Myo3b-ARB1 complex was solved by molecular replacement with the model of short ANK repeats (1N0R) using PHASER (*Mccoy et al., 2007*). Structure of the Espin1-AR/Myo3b-ARB2 complex was also solved by molecular replacement with the 10 ANK repeats of Espin1 in the previous structure as the search model. Phases were greatly improved after auto-building by Buccaneer (*Cowtan, 2006*). Further manual model building and refinement were completed iteratively using COOT (*Emsley et al., 2010*) and PHENIX (*Adams et al., 2010*). The final model was validated by MolProbity (*Chen et al., 2010*). The final refinement statistics are summarized in *Table 1*. All structure figures were prepared by PyMOL (http://www.pymol.org). The coordinates of the structures reported in this work have been deposited to PDB under the access codes of 5ET1 and 5ET0 for the Espin1-AR/Myo3b-ARB1 and Espin1-AR/Myo3b-ARB2 structures, respectively.

### Isothermal titration calorimetry assay

Isothermal titration calorimetry (ITC) measurements were carried out on a MicroCal iTC$_{200}$ at 25°C, except for the two endothermic titrations which were performed at 16°C. Titration buffer contained

50 mM Tris-HCl pH 7.8, 1 mM DTT, 1 mM EDTA and 200 mM NaCl. Each titration point was performed by injecting a 2 µL aliquot of a protein sample from a syringe into a protein sample in the cell at a time interval of 120 s to ensure that the titration peak returned to the baseline. The titration data were analyzed by Origin7.0 (Microcal).

## COS7 cell culture and transfection

COS7 cells were cultured in Dulbecco's Modified Eagle Medium (Corning) supplemented with 1 mM Sodium Pyruvate, 4 mM L-glutamine, 4.5 g/L D-Glucose, 10% fetal bovine serum (FBS) (Gemini), and 100 units of penicillin-streptomycin (Corning). Cultured COS7 cells were maintained at 37°C with 5% $CO_2$ in air. For transfections, $30–40 \times 10^3$ cells were plated on acid washed 22 mm square 1.5# glass coverslips and allowed to adhere over-night. 24 hr later, cells were transiently transfected using FUGENE HD transfection reagent (Promega) as per manufacturer's protocol and imaged after ~20–30 hr.

## Live-cell imaging of COS-7 cells

For live cell imaging, the coverslips with transfected cells were placed in rose chambers filled with Opti-MEM media without phenol red (Life Technologies) and supplemented with 5% FBS (Gemini) and 100 units of Penicillin-streptomycin. Images were acquired by using a TE2000-PFS fluorescence microscope (Nikon Instruments) with a 60x/1.4 N.A. phase objective. Image acquisition was managed by NIS-Elements AR (Nikon Instruments) and the tip to call body measurements were done using ImageJ (ND2 plugin), as described previously (*Quintero et al., 2010*). Data are expressed as mean ± SEM. The mutant groups were compared with the wild type groups by two-tailed student's *t* test.

## Actin bundling

Rabbit skeletal muscle actin (Cytoskeleton) were hydrated in 5 mM Tris-HCl, 0.2 mM $CaCl_2$, 0.2 mM ATP, 0.5 mM DTT, pH 8.0, on ice for 1 hr and centrifuged at 150,000 g for 10 min at 4°C. Actin concentration in supernatant was determined by NanoDrop. Actin was polymerized at room temperature for 1 hr after adding one-tenth volume of 10×polymerizing buffer (500 mM KCl, 20 mM $MgCl_2$, 10 mM ATP). Bundles were prepared by mixing 5 µM F-actin with 1.25 µM Espin1 and incubating at room temperature for 1 hr, with or without 1.25 µM Myo3a-ARB1/2 or 0.625 µM Myo3a-ARB12 added in the mixtures. For fluorescence microscopy, F-actin were labeled with Alexa Fluor 555 Phalloidin. Aliquots (5 µl) were delivered onto microscope slides. The cover slips were then placed over the drop of samples gently. All the samples were imaged using a Fixed Stage Upright Microscope (Olympus). Samples for TEM (FEI Tecnai 20) were adsorbed to glow-discharged, carbon-coated formvar films on copper grids for 1 min and negatively stained with 0.75% (m/v) uranium formate for 30 s.

## Acknowledgements

We thank the Shanghai Synchrotron Radiation Facility (SSRF) BL17U and BL19U for X-ray beam times. We also thank the Center for Biological Imaging (CBI) of the Institute of Biophysics, Chinese Academy of Science for use of the electron microscopy facility and Dr. Jun Ma in Prof. Xinzheng Zhang's lab for his help of taking EM images. This work was supported by grants from a 973 program grant from the Minister of Science and Technology of China (2014CB910204) and from RGC of Hong Kong (663811, 663812, and AoE-M09-12) to MZ, a 973 program grant (2014CB910202) to WF, and National Natural Science Foundation of China (No.31400647) and Guangdong Natural Science Foundation (No. S2012010008170) grants to WL. MZ is a Kerry Holdings Professor in Science and a Senior Fellow of IAS at HKUST.

## Additional information

### Competing interests

MZ: Reviewing editor, *eLife*. The other authors declare that no competing interests exist.

## Funding

| Funder | Grant reference number | Author |
| --- | --- | --- |
| Research Grants Council, University Grants Committee | 663811, 663812, and AoE-M09-12 | Mingjie Zhang |
| Ministry of Science and Technology of the People's Republic of China | 2014CB910204 | Mingjie Zhang |
| National Natural Science Foundation of China | No.31400647 | Wei Liu |
| Ministry of Science and Technology of the People's Republic of China | 2014CB910202 | Wei Feng |

The funders had no role in study design, data collection and interpretation, or the decision to submit the work for publication.

## Author contributions

HL, JL, Conception and design, Acquisition of data, Analysis and interpretation of data, Drafting or revising the article; MHR, Acquisition of data, Analysis and interpretation of data, Drafting or revising the article; NY, XD, SN, Acquisition of data, Analysis and interpretation of data; QL, WF, Conception and design, Analysis and interpretation of data; JW, CMY, Analysis and interpretation of data, Drafting or revising the article; WL, MZ, Conception and design, Analysis and interpretation of data, Drafting or revising the article

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
