## [Decision Letter]

Thank you for submitting your work entitled "Myosin III-mediated Dimerization and Stimulation of Actin Bundling Activity of Espin" for consideration by *eLife*. Your article has been favorably evaluated by John Kuriyan (Senior editor) and three reviewers, one of whom, Cynthia Wolberger, is a member of our Board of Reviewing Editors, and another is James Bartles.

The reviewers have discussed the reviews with one another and the Reviewing Editor has drafted this decision to help you prepare a revised submission.

This manuscript sheds light on the molecular details underlying the way in which Myo3 binds to Espin1 to regulate actin bundle assembly in hair cells. Espin1 contains a set of ankyrin repeats that bind to Myo3. The authors identify two related sequences within Myo3, ARB1 and ARB2, and show that these sequences bind independently to the ankyrin repeat domain (ARD) of Espin1. Structures of the Espin1 ankyrin repeats bound to ARB1 and to ARB2 peptides from Myo3 show that each binds similarly to the ARD, suggesting that a single Myo3 can trigger dimerization of Espin1. The authors also provide structural and biophysical evidence that the Espin1 autoinhibitory sequence, which inhibits Espin1 binding to actin, resembles the two Myo3 ARB sequences and competes for a common site in the ankyrin repeats. Studies in hair cells support a role for the ARB1/2-ARD interaction in colocalization of Myo3 and Espin1, while light microscopy and electron micrographs provide evidence that Espin1 mediates higher order bundling of actin.

Review summary:

All three reviewers were enthusiastic about this study and its combination of approaches that provide fresh insights into Myo3-Espin1 interactions in hair cells. There were several concerns raised by the reviewers, one of which will require further experiments, with the remainder most likely addressable using available data and through revision of the figures and text.

Essential revisions:

1) The ITC data showing a lower affinity of Myo3a-ARB1 for full-length Espin1 or a fragment extending through the inhibitory domain are consistent with an autoinhibition model. However, these data do not show that it is the putative AI domain, as opposed to some other region of Espin1, that is responsible for the elevated K_d_ and switch from exothermic to endothermic binding reaction. The authors should assay binding of ARB either to an Espin1 fragment truncated just before the AI sequence (and no more) or introduce mutations in AI domain residues that relieve autoinhibition, and then show that Myo3a binds more tightly to this Espin1 fragment.

2) The authors state that ARB1 and ARB2 bind in a similar manner to the ankyrin repeats, yet the cartoon/ribbons figures in panels 2A and 2D appear to show differences that could reflect somewhat different contacts, at least for a subset of the residues. There should be a detailed comparison of these two structures in figures that would enable the reader to do a side-by side comparison of contacts by ARB1 versus ARB2. Differences in the way in which the two bind to the AR may underlie the apparent differences that mutations in ARB1 versus ARB2 have in the in vivo experiments, where mutations of the Tyr residues in ARB2 seem to have a much stronger effect on recruitment (Figure 5). It may be helpful to show electron density for the ARB2 structure as well, either to justify the chain tracing or show where the peptide trace may be ambiguous (and therefore may account for the apparent difference).

3) The schematic (Figure 6) which shows parallel pairs of actin filaments that are not cross-linked, and no involvement of xAB or the ABM of Myo3.

4) One issue that needs to be discussed further in the context of the proposed mechanism is the relatively low levels of these two proteins in the stereocilium. For chick utricle, the data are listed in the Shin et al. 2013 Nature Neuroscience paper, which is included in the current list of references.

5) The authors should explicitly acknowledge the different methodology used in the current manuscript, which uses post-polymerization bundling, and the previous study by Zheng et al. (2014), which used co-polymerization bundling to examine Espin 1 autoinhibition and the regulation of the Espin 1 xAB domain by the MYO3 peptide.

6) The use of the term "dimerization" is misleading and should be replaced with another term that more accurately describes the fact that the two adjacent ARB motifs, each of which recruit an Espin, clusters two Espins close to one another.

---

## [Author Response]

Essential revisions:

*1) The ITC data showing a lower affinity of Myo3a-ARB1 for full-length Espin1 or a fragment extending through the inhibitory domain are consistent with an autoinhibition model. However, these data do not show that it is the putative AI domain, as opposed to some other region of Espin1, that is responsible for the elevated K_d_ and switch from exothermic to endothermic binding reaction. The authors should assay binding of ARB either to an Espin1 fragment truncated just before the AI sequence (and no more) or introduce mutations in AI domain residues that relieve autoinhibition, and then show that Myo3a binds more tightly to this Espin1 fragment.*

Thanks for the great suggestions. We have performed ITC titration experiments between Myo3a-ARB1/2 and Espin1-1-494 (truncated right before the AI sequence). Consistent with our prediction, the affinities are in a similar range with that between Myo3a-ARB1/2 and Espin1-AR. The bindings of ARB1/2 to Espin1-1-494 also become exothermic. We have integrated the ITC data between Myo3a-ARB1 and Espin1-1-494 into Figure 4 (as Figure 4), and attached the ITC data between Myo3a-ARB2 and Espin1-1-494 (Figure 7) for your reference.

Author response image 1.ITC results showing that Myo3a-ARB2 binds to Espin1-1-494 with similar affinity compared to Espin1-AR.**DOI:**
http://dx.doi.org/10.7554/eLife.12856.019

2) The authors state that ARB1 and ARB2 bind in a similar manner to the ankyrin repeats, yet the cartoon/ribbons figures in panels 2A and 2D appear to show differences that could reflect somewhat different contacts, at least for a subset of the residues. There should be a detailed comparison of these two structures in figures that would enable the reader to do a side-by side comparison of contacts by ARB1 versus ARB2. Differences in the way in which the two bind to the AR may underlie the apparent differences that mutations in ARB1 versus ARB2 have in the in vivo experiments, where mutations of the Tyr residues in ARB2 seem to have a much stronger effect on recruitment (Figure 5). It may be helpful to show electron density for the ARB2 structure as well, either to justify the chain tracing or show where the peptide trace may be ambiguous (and therefore may account for the apparent difference).

Following the reviewers’/editor’s suggestion, we have added a paragraph and a supplementary figure (Figure 3—figure supplement 2) discussing the differences between Myo3b-ARB1 and Myo3b-ARB2 in the revised manuscript:

“Comparing the structures of Espin1-AR in complex with Myo3b-ARB1 and Myo3b-ARB2, the “YY” motif and the “KxL” motif are essentially in the same places (Figure 3—figure supplement 2B and C). Despite the high similarity, there are still a few minor differences. First of all, the C-terminal α-helix of ARB2 is shorter. The interaction is mediated by a hydrogen bond between Asp1294 and Tyr144, instead of the more extensive interaction observed in Myo3b-ARB1 complex (Figure 3—figure supplement 2B). Moreover, the positively charged residues in the N-terminus of ARB2 cannot be reliably built, probably due to the high salt concentration in the crystallization buffer (1.6 M ammonium sulfate). Nonetheless, clear electron density can be observed near the negatively charged surface (Figure 3—figure supplement 2A). Indeed, substitutions of these positively charged residues with Ala weakened the binding (Figure 3 and Figure 3—figure supplement 3G-I). Furthermore, the involvement of more positively charged residues of ARB2 may compensate for the less extensive interaction in its shorter C-terminal helix, thus resulting in a similar binding affinity to Espin1-AR as ARB1 does (107 nM for ARB1 vs 53 nM for ARB2, Figure 1).”

3) The schematic (Figure 6) which shows parallel pairs of actin filaments that are not cross-linked, and no involvement of xAB or the ABM of Myo3.

Thanks for pointing this out. We have modified the schematic representation to include the involvement of xAB and ABM of Myo3a in the revised Figure 6. We have also added a sentence in the legend of Figure 6, which reads as “The xAB region of this Myo3-activated Espin1 and the ABM of Myo3a may further stabilize the higher order actin bundles”.

4) One issue that needs to be discussed further in the context of the proposed mechanism is the relatively low levels of these two proteins in the stereocilium. For chick utricle, the data are listed in the Shin et al. 2013 Nature Neuroscience paper, which is included in the current list of references.

The reviewers/editors have raised a very interesting point. The protein level for Espin in stereocilia is relatively low when comparing to fascin and plastin, two additional actin filament crosslinking proteins. It is unlikely that Espin plays a major structural role in the cross-linking of F-actin in stereocilia. When reading the Nature Neuroscience paper by Shin et al. 2013, it is interesting to note that the protein level for Myo3a/b together is at a similar range as Espin (a few hundred copies per stereocilium). What we propose here (as also proposed by others) is that Myo3 and Espin (perhaps also espin-like protein) might work together to perform certain regulating role in elongation and widening of actin bundles. We have expanded our Discussion in the revised manuscript by adding the following:

“It has been proposed that Espin may play a regulating role in elongation and widening of actin bundle by concentrating at the tip of stereocilia instead of major structural cross-linking roles (Avenarius et al., 2014; Loomis et al., 2003). Interestingly, the abundance of Myo3 (both Myo3a and Myo3b together) is similar to that of Espin (a few hundred copies per stereocilium) (Shin et al., 2013), indicating that Myo3 and Espin1 may work together to regulate higher order actin bundle structure formation and stability in stereocilia.”

*5) The authors should explicitly acknowledge the different methodology used in the current manuscript, which uses post-polymerization bundling, and the previous study by Zheng et al. (2014), which used co-polymerization bundling to examine Espin 1 autoinhibition and the regulation of the Espin 1 xAB domain by the MYO3 peptide.*

Thanks for pointing this out. We have modified the manuscript to acknowledge the different methodology in the current manuscript as follows:

“By adding ARB1 or ARB2 to Espin1 and F-actin containing solution, some of the actin bundles were cross-linked and formed clusters under FM and TEM (Figure 6). Similar phenomena have also been observed earlier using Myo3/Espin1/actin co-polymerization bundling assay (Zheng et al., 2014), instead of the post-polymerization bundling assay employed in this study.”

6) The use of the term "dimerization" is misleading and should be replaced with another term that more accurately describes the fact that the two adjacent ARB motifs, each of which recruit an Espin, clusters two Espins close to one another.

Thanks for alerting us on this point. We now use the term “cross-linking” which should be more accurate in describing assembly of two Espin1 molecules by the two adjacent ARB motifs of Myo3.